# DHPA Protects SH-SY5Y Cells from Oxidative Stress-Induced Apoptosis via Mitochondria Apoptosis and the Keap1/Nrf2/HO-1 Signaling Pathway

**DOI:** 10.3390/antiox11091794

**Published:** 2022-09-12

**Authors:** Yunhui Cai, Ran Xiao, Yadan Zhang, Diya Xu, Ni Wang, Mengze Han, Yili Zhang, Lin Zhang, Wenhua Zhou

**Affiliations:** 1Hunan Key Laboratory of Processed Food for Special Medical Purpose, Hunan Key Laboratory of Forestry Edible Resources Safety and Processing, School of Food Science and Engineering, Center South University of Forestry and Technology, Changsha 410004, China; 2National Engineering Research Center of Rice and Byproduct Deep Processing, Center South University of Forestry and Technology, Changsha 410004, China; 3SINOCARE Inc., Changsha 410004, China

**Keywords:** Alzheimer’s disease, oxidative stress, SH-SY5Y cell, mitochondria, Nrf2, ROS

## Abstract

Oxidative stress in the brain is highly related to the pathogenesis of Alzheimer’s disease (AD). It could be induced by the overproduction of reactive oxygen species (ROS), produced by the amyloid beta (Aβ) peptide and excess copper (Cu) in senile plaques and cellular species, such as ascorbic acid (AA) and O_2_. In this study, the protective effect of 5-hydroxy-7-(4′-hydroxy-3′-methoxyphenyl)-1-phenyl-3-heptanone (DHPA) on Aβ_(1–42)_/Cu^2+^/AA mixture-treated SH-SY5Y cells was investigated via in vitro and in silico studies. The results showed that DHPA could inhibit Aβ/Cu^2+^/AA-induced SH-SY5Y apoptosis, OH· production, intracellular ROS accumulation, and malondialdehyde (MDA) production. Further research demonstrated that DHPA could decrease the ratio of Bax/Bcl-2 and repress the increase of mitochondrial membrane potential (MMP) of SH-SY5Y cells, to further suppress the activation of caspase-3, and inhibit cell apoptosis. Meanwhile, DHPA could inhibit the Aβ/Cu^2+^/AA-induced phosphorylation of Erk1/2 and P38 in SH-SY5Y cells, and increase the expression of P-AKT. Furthermore, DHPA could bind to Keap1 to promote the separation of Nrf2 to Keap1 and activate the Keap1/Nrf2/HO-1 signaling pathway to increase the expression of heme oxygenase-1 (HO-1), quinone oxidoreductase-1 (NQO1), glutathione (GSH), and superoxide dismutase (SOD). Thus, our results demonstrated that DHPA could inhibit Aβ/Cu^2+^/AA-induced SH-SY5Y apoptosis via scavenging OH·, inhibit mitochondria apoptosis, and activate the Keap1/Nrf2/HO-1 signaling pathway.

## 1. Introduction

Alzheimer’s disease (AD) is a neurodegenerative disease that causes memory loss, cognitive impairment, and behavioral dysfunction [1]. In AD brains, high concentrations of Cu^2+^ are detected (about 0.4 mM) in amyloid beta (Aβ)-accumulated senile plaques [2,3]. Moreover, the Aβ–Cu^2+^ complex could react with ascorbic acid (AA) (500 μM–10 mM) in the brain to continuously generate excessive hydroxyl radicals (OH·) and reactive oxygen species (ROS) via Fenton and Haber–Weiss reactions [4,5]. These ROS radicals will further induce oxidative damage to neurons and aggravate the progression of AD [6].

Mitochondria are important sources of ROS [7,8]. Excess ROS could influence the expression of B-cell lymphoma 2 (Bcl-2) and Bcl2-associated X (Bax), changing the mitochondrial membrane potential (MMP), inducing the expression of caspase-3 (cleaved), and inducing neuronal apoptosis [9].

The Kelch-like ECH-associated protein 1-nuclear factor-E2-related factor 2 (Keap1-Nrf2) signaling pathway, together with the antioxidant response element (ARE), is an important pathway for cells to resist oxidative stress [10]. Under normal physiological conditions, Nrf2 binds to Keap1 in an inactive state in the cytoplasm. If it has not been activated, Nrf2 will be ubiquitinated and then degraded [10,11]. When the cells are stimulated by internal or external oxidative stress and electrophilic or chemical substances, Nrf2 is activated [12]. There are two main ways to activate Nrf2. One is to mutate two cysteine residues, Cys273 and Cys288, in the intervening region (IVR) of Keap1, resulting in reduced Keap1-dependent ubiquitination of Nrf2, increased intracellular stability, and translocation of Nrf2 into the nucleus [13,14]. The other is to phosphorylate Nrf2 through multiple protein kinase pathways, such as mitogen-activated protein kinases (MAPKs), phosphatidylinositol-3-kinase (PI3K), and protein kinase C (PKC) to separate it from Keap1 [12]. After activation, Nrf2 is released and transferred to the nucleus [15], and the ARE promotes the expressions of antioxidant enzymes, such as heme oxygenase-1 (HO-1) and quinone oxidoreductase-1 (NQO1) [16]. Meanwhile, Keap1-Nrf2 can also induce the decomposition of superoxide and superoxide by inducing the expressions of superoxide dismutase (SOD), superoxide reductase, glutathione peroxidase, etc. [17,18].

As Keap1 plays an important role in the ubiquitination and degradation of Nrf2, it is commonly considered the target for molecular docking (MD) simulations to interrupt the Keap1/Nrf2 protein interactions and activate Nrf2 antioxidative signaling [19]. Several natural compounds have been reported to activate Nrf2 via in vitro, in vivo, and MD simulation experiments [20,21]. Research shows that sargahydroquinoic acid, which is extracted from algae, could bind to Keap1 via hydrogen bonds and van der Waals forces, and induce Nrf2 activation (to mediate the antioxidant defense system in Aβ-induced oxidative stress) [22]. Phlorizin, which is extracted from apple peel, is also reported to bind to the Keap1 protein to separate the Nrf2 protein from Keap1 and activate Nrf2 (to enter the nucleus and promote the expressions of antioxidant-related genes) [23].

*Alpinia officinarum Hance* is a traditional medicine in China; it mainly contains diarylheptanoids, flavonoids, essential oils, and other bioactive components [24]. It has been reported that diarylheptane (DAH) extracted from *Alpinia officinarum Hance* has various biological activities, such as anticancer–antibacterial [25], anti-inflammatory, and antioxidant properties [26,27]. Moreover, 5-hydroxy-7-(4′-hydroxy-3′-methoxyphenyl)-1-phenyl-3-heptanone (DHPA), is one of the DAHs extracted from the rhizome of *Alpinia officinarum Hance* [28]. It can be used as a pancreatic lipase inhibitor [29]; it also has anti-*Helicobacter pylori* [30] and antitubercular abilities. Meanwhile, it has been reported that DHPA has significant antioxidant and anti-inflammatory abilities [27], but the study on AD is rare. In this study, the protective effect of DHPA on SH-SY5Y cells against Aβ_(1–42)_/Cu^2+^/AA-induced oxidative stress was investigated. The potential mechanism was also studied with in vitro and in silico methods.

## 2. Materials and Methods

### 2.1. Materials and Solution Preparation

Materials: 5-hydroxy-7-(4′-hydroxy-3′-methoxyphenyl)-1-phenyl-3-heptanone (DHPA) (purity ≥ 98%) was purchased from Shanghai Yansheng Industrial Co., Ltd. (Shanghai, China). Dulbecco’s Modified Eagle Media: Nutrient Mixture F-12 (DMEM/F12) (*v*:*v*, 1:1) cell culture medium (cat. no. C11330500BT), fetal bovine serum (FBS) (cat. no. 10270-106), and 0.25% trypsin (cat. no. 25200-056) were purchased from Gibco Co. (Grand Island, NY, USA). Aβ_(1–42)_ (cat. no. 107761-42-2) was purchased from the American Peptide Company Co., Inc. (Sunnyvale, CA, USA). Annexin V-FITC(PI) (cat. no. AO2001-02P-H) Apoptosis Analysis Kit was obtained from Sungene Biotech Co., Ltd. (Tianjin, China). BeyoECL Plus (cat. no. P0018S), active oxygen detection kit (cat. no. S0033S), and mitochondrial membrane potential detection kit JC-1 (cat. no. C2006) were obtained from Beyotime Biological Co., Ltd. (Shanghai, China). The BCA protein quantification assay kit (cat. no. P001B-1) was obtained from Labgic Technology Co., Ltd. (Hefei, China). Coumarin-3-carboxylic acid (CCA) (cat. no. C85603) and 3-(4,5-dimethylthiazol-2-yl)-2,5-diphenyltetrazolium bromide (MTT) (cat. no. M2003), CuSO_4_ (cat. no. 209198) were purchased from Sigma Chemical Co. (St. Louis, MO, USA). Ascorbic acid (cat. no. A800296) and NaOH (cat. no. 10019718) and other reagents were purchased from Sinopharm Chemical Reagent Co., Ltd. (Beijing, China).

Bcl2 (1:1000, cat. no. A19693), Bax (1:1000, cat. no. A12009), Nrf2 (1:1000, cat. no. A0674), Keap1 (1:1000, cat. no. A1820), HO-1 (1:1000, cat. no. A1346), and NQO1 (1:1000, cat. no. A0047) were bought from ABclonal Technology Co., Ltd. (Wuhan, Hubei, China). Cleaved caspase-3 (1:1000, cat. no. ARG57512) and P-P38 (1:500, cat. no. ARG51850) were obtained from Arigo Biolaboratories (Hsinchu City, Taiwan, China). P38 (1:1000, cat. no. 9212), Erk1/2 (1:1000, cat. no. 4695), P-Erk1/2 (1:2000, cat. no. 4370), AKT (1:1000, cat. no. 9272), and P-AKT (1:2000, cat. no. 4060) were bought from Cell Signaling Technology, Inc. (Danvers, MA, USA). Additionally, malondialdehyde (MDA; cat. no. A003-4-1), GSH (cat. no. A061-1-2), and superoxide dismutase (SOD; cat. no. A001-3-2) test kits were bought from Nanjing Jiancheng Bioengineering Institute (Nanjing, China).

Aβ_(1–42)_ stock was freshly prepared by dissolving Aβ_(1–42)_ (0.5 mg) in 20 mM NaOH (220 μL) solution. Copper stock was prepared by dissolving CuSO_4_ (0.25 g) in 100 mM H_2_SO_4_ (10 mL), diluted with 1 mM H_2_SO_4_ to 1 mM CuSO_4_, and diluted with deionized (DI) water (100 μM). Solutions of AA (10 mM) were freshly prepared with DI water (for the CCA assay) or complete media (cell assays). The DHPA (2 mM) stock was prepared with dimethyl sulfoxide (DMSO). To prepare the Aβ_(1–42)_ (10 μM)/Cu^2+^ (5 μM)/AA (1 mM)/DHPA mixture, different stock volumes were added into the DI water (for the CCA assay) or complete media (cell assays) to obtain the desired concentrations. The final concentration of NaOH was 0.4 mmol/L, and the final concentration of DMSO was less than 0.1%. The control group solution was prepared with the same formula as the experimental group, except Aβ_(1–42)_, Cu^2+^, AA, or DHPA.

### 2.2. Cell Culture

Human neuroblastoma SH-SY5Y cells were purchased from the American Type Culture Collection (Washington, DC, USA). SH-SY5Y cells were routinely grown in Dulbecco’s Modified Eagle Medium supplemented with 10% fetal bovine serum, 2 mM L-glutamine, and 1% streptomycin/penicillin–streptomycin at 37 °C in a humidified incubator (S111, Thermo Fisher Scientific, Waltham, MA, USA) with 5% CO_2_.

### 2.3. Cell Viability Assay (MTT Assay)

When SH-SY5Y cells grew to the log phase, they were digested with trypsin for counting, the cell densities were adjusted to 1 × 10^4^ cells/mL (100 μL), and added to a 96-well plate, 100 μL per well. Then, the plated 96-well plate was placed in an incubator for 6 h. Afterward, the cells were stably adhered to the wall, cultured with Aβ_(1–42)_ (10 μM)/Cu^2+^ (5 μM)/AA (1 mM) mixtures in the absence or presence of different concentrations of DHPA (0, 0.5, 2, 4, 6, 8, 10 μM) for 24 h; 100 μL of 10% MTT was added to each well, incubated for 4 h, were carefully removed, and 100 μL of DMSO was added. The absorbance value was detected at 490 nm with an automated microplate spectrophotometer (Spectra Max i3X, Sunnyvale, CA, USA). Cell viability is expressed as a percentage of the control.

### 2.4. Morphological Observations

SH-SY5Y cells in the logarithmic growth at a density of 1 × 10^4^ cells/mL (1 mL) were cultured onto a 6-well plate for 6 h. Then SH-SY5Y cells were cultured with Aβ_(1–42)_ (10 μM)/Cu^2+^ (5 μM)/AA (1 mM) mixtures in the absence or presence of different concentrations of DHPA (0, 0.5, 4, 10 μM) for 24 h. Cell morphology was obtained and images were captured with an inverted fluorescence microscope (Ti2-E, Tokyo, Japan).

### 2.5. Flow Cytometry Assay

SH-SY5Y cells (1 × 10^4^ cells/mL, 1 mL) were placed on a 6-well plate for 6 h, and treated with Aβ_(1–42)_ (10 μM)/Cu^2+^ (5 μM)/AA (1 mM) mixtures in the absence or presence of different concentrations of DHPA (0, 0.5, 4, 10 μM) for 24 h at 37 °C. Cell apoptosis was detected with the Annexin V-FITC(PI) Apoptosis Analysis kit. The cells were washed and centrifuged 3 times, resuspended in a flow tube, and incubated with fluorescein isothiocyanate-(FITC-)-coupled annexin V (AV) and propidium iodide (PI) solution for 15 min. The fluorescence signal was detected with a NovoCyte flow cytometer (Agilent Technologies Inc., Palo Alto, CA, USA) and analyzed by NovoExpress software.

### 2.6. Detection of Hydroxyl Radical

CCA was used as a fluorescence probe for OH· determination [31] with a Hitachi F-4600 spectrofluorometer from Hitachi High-Tech Corporation (Tokyo, Japan). CCA fluorescence was recorded at 540 nm, with an excitation wavelength of 390 nm. The widths of the entrance and exit slits were both 5 nm. The OH· amounts in the solutions of the Aβ_(1–42)_ (10 μM)/Cu^2+^ (5 μM)/AA (1 mM) mixtures in different concentrations of DHPA (0.5, 4, 10 μM) were detected with a CCA probe. The fluorescence ratio was calculated as follows:Ratio (%) = 100 × F/F_0_(1)
where F is the CCA fluorescence intensity in each solution and F_0_ is the CCA fluorescence intensity of the Aβ_(1–42)_ (10 μM)/Cu^2+^ (5 μM)/AA (1 mM) mixture.

### 2.7. Intracellular Reactive Oxygen Species Determination

SH-SY5Y cells (1 × 10^4^ cells/mL, 1 mL) were placed on a 6-well plate for 6 h, and then treated with Aβ_(1–42)_ (10 μM)/Cu^2+^ (5 μM)/AA (1 mM) at different concentrations of DHPA (0, 0.5, 4, 10 μM) for 24 h at 37 °C, and then were treated according to the DCFH-DA kit instructions. The fluorescent pictures were taken with an inverted fluorescence microscope (Ti2-E, Tokyo, Japan). The fluorescence intensity was recorded with the Hitachi F-4600 spectrofluorometer from Hitachi High-Tech Corporation (Tokyo, Japan) with the excitation and emission wavelengths at 485 and 525 nm. The fluorescence ratio was calculated as follows:Ratio (%) = 100 × F/F_0_(2)
where F is the DCFH-DA fluorescence intensity of Aβ_(1–42)_ (10 μM)/Cu^2+^ (5 μM)/AA (1 mM)/DHPA (0.5, 4, 10 μM) mixture-treated SH-SY5Y cells and F_0_ is the DCFH-DA fluorescence intensity of the Aβ_(1–42)_ (10 μM)/Cu^2+^ (5 μM)/AA (1 mM)-treated SH-SY5Y cells.

### 2.8. Intracellular SOD, GSH, and MDA Content Determination

SH-SY5Y cells (1 × 10^4^ cells/mL, 1 mL) were grown on a 6-well plate for 6 h and then treated with Aβ_(1–42)_ (10 μM)/Cu^2+^ (5 μM)/AA (1 mM) at different concentrations of DHPA (0, 0.5, 4, 10 μM) for 24 h at 37 °C. Glutathione (GSH), superoxide dismutase (SOD), and malondialdehyde (MDA) were measured following the kit instructions (Nanjing Jiancheng Bioengineering Institute, Nanjing, China). The absorbance value was detected with a microplate reader.

### 2.9. Mitochondrial Membrane Potential Determination

A fluorescent probe 5,5′,6,6′-tetrachloro-1,1′,3,3′-tetraethyl-imidacarbocyanine (JC-1) was used to detect mitochondrial membrane potential (MMP) of SH-SY5Y cells. The cell treatment method is the same as the description in Section 2.7. The cells were treated with JC-1 dye following the JC-1 kit instructions. The fluorescence intensity was recorded with a microplate reader and the MMP was obtained as the ratio of the JC-1 monomer (FI 530)/JC-1 aggregate (FI 590) fluorescence intensity.

### 2.10. Western Blot Analysis

Cytosolic protein and nuclear protein samples were collected from DHPA (0, 0.5, 4, 10 μM) and Aβ_(1–42)_ (10 μM)/Cu^2+^ (5 μM)/AA (1 mM)-treated cells. Protein levels were measured with a BCA protein quantification assay kit (Labgic Technology Co., Ltd. (Hefei, China). Proteins were separated by polyacrylamide/sodium dodecyl sulfate gel electrophoresis and transferred onto polyvinylidene fluoride membranes. Membranes were blocked with 5% bovine albumin in Tris-buffered saline containing 0.05% Tween 20 (TBST), followed by incubation with the respective primary antibodies diluted in 5% (M/V) bovine albumin overnight at 4 °C. After washing with Tris-buffered saline with 0.05% Tween 20 (TBST), the membranes were incubated with an HRP-secondary antibody for 1 h. Immunoreactive bands were visualized with a chemiluminescent substrate kit and imaged with Gel Imager 721-BR10883 from BIO-RAD Co., Ltd. (Hercules, CA, USA). Bands were detected using Image-J software.

### 2.11. Molecular Docking

The interaction between the DHPA and Keap1 protein (PDB:1X2J, origin: *Mus musculus*) was investigated with a similar method reported by Liu and co-workers [23]. The molecular structure of the Keap1 protein was downloaded from the Protein Data Bank (http://www.rcsb.org (accessed on 23 May 2022)). The molecular structures of DHPA and curcumin were downloaded from http://www.chemicalbook.com (accessed on 23 May 2022). The molecular docking experiments were performed with AutoDock Vina. Moreover, the binding energies and interaction sites were analyzed by Discover Studio 2019.

### 2.12. Statistical Analysis

The results were calculated by using the SPSS 22.0 statistical analysis, and Origin 2018 was used for drawing. Adobe Illustrator software was used to combine graphics and draw a schematic diagram of the mechanisms. Each experiment and each group of data were repeated three times, and the results are expressed by mean ± standard deviation (mean ± SD).

## 3. Results

### 3.1. Effects of DHPA on Aβ_(1–42)_/Cu^2+^/AA-Treated SH-SY5Y Cells Viability

Amyloid beta (Aβ) peptides aggregated with redox-active metal ions (such as Cu^2+^ and Fe^3+^) have been found to exist in senile plaques in the brains of AD patients [32]. These metal ions can easily bind Aβ peptides to form complexes [32,33] and react with substances such as ascorbic acid (AA) and O_2_ in cells to promote the production of OH [33]. Thus, in this study, the Aβ_(1–42)_/Cu^2+^/AA system was used as the OH production model to simulate the OH-producing process in the senile plaques of AD patients.

To investigate the cytotoxicity of the 5-hydroxy-7-(4′-hydroxy-3′-methoxyphenyl)-1-phenyl-3-heptanone (DHPA) to SH-SY5Y neuroblastoma cells, the cell viabilities were detected with the MTT assay. When the concentration of DHPA was at 10 μM, the cell viability was 89.9%. When the concentration of DHPA increased, the cell viability decreased. Moreover, when the concentration of DHPA was at 40 μM, the cell viability decreased to 65.7% (Figure 1B). As shown in Figure 1C, concentrations of Cu^2+^ (5 μM) and AA (1 mM) at lower levels showed almost no cell toxicity to SH-SY5Y cells. Aβ_(1–42)_ (10 μM) reduced the cell viability to 92.3% (Figure 1C). However, when the same concentrations of Aβ_(1–42)_, Cu^2+^, and AA were mixed, the cell viability decreased to 52.2%. This may have been due to the oxidative damage from OH·, which was produced by the Aβ_(1–42)_/Cu^2+^/AA system, and the cell toxicities from Aβ_(1–42)_ toxic aggregates.

To investigate the protective effect of DHPA on the Aβ_(1–42)_/Cu^2+^/AA mixture-treated SH-SY5Y cells, the cells were incubated with Aβ_(1–42)_/Cu^2+^/AA/DHPA mixtures. As shown in Figure 1D, when the DHPA concentration increased from 0 to 10 μM, the cell viability of the Aβ_(1–42)_/Cu^2+^/AA-treated SH-SY5Y cells increased from 52.2% to 87.6%. Moreover, when the concentration of DHPA increased to 20 μM, the cell viability decreased to 79.6%. These results indicated that a concentration of DHPA below 10 μM could inhibit the Aβ_(1–42)_/Cu^2+^/AA system-induced cytotoxicity.

### 3.2. Effects of DHPA on Aβ_(1–42)_/Cu^2+^/AA-Induced SH-SY5Y Cell Apoptosis

As shown in Figure 2A, cells in the control group (untreated cells) showed normal cell morphology, were well-adhered, and had clear edges. The cells in the Aβ_(1–42)_/Cu^2+^/AA-treated group showed ‘shrunken atrophy’ and blurred ruptured boundaries; most cells were apoptotic. When the cells were treated with DHPA, the morphological damage and density reduction caused by Aβ_(1–42)_/Cu^2+^/AA were restored in a DHPA dose-dependent manner. The shapes of the cells gradually became normal, and the edges were gradually clear, but a small number of suspended cells could still be observed. Annexin V/PI staining was also used to verify the effect of DHPA on Aβ_(1–42)_/Cu^2+^/AA-induced cell apoptosis. As shown in Figure 2B, with the addition of DHPA, the apoptosis rate of Aβ_(1–42)_/Cu^2+^/AA-treated SH-SY5Y cells decreased from 47.23% to 22.41%, which indicated that DHPA could inhibit Aβ_(1–42)_/Cu^2+^/AA-induced cell apoptosis in a dose-dependent manner.

### 3.3. Scavenging Effect of DHPA on Aβ_(1–42)_/Cu^2+^/AA-Produced OH

To investigate the reason behind the inhibition effect of DHPA on Aβ_(1–42)_/Cu^2+^/AA-induced cell apoptosis, the scavenging effect of DHPA on OH·, produced by the Aβ_(1–42)_/Cu^2+^/AA mixture, was detected with the CCA method. With the increase in the DHPA concentration (0, 0.5, 4, 10 μM), the amount of OH· decreased from 100% to 54.82% (Figure 3). The results indicate that DHPA could inhibit the production of OH· by the Aβ_(1–42)_/Cu^2+^/AA mixture in a dose-dependent manner.

### 3.4. Effect of DHPA on the Accumulation of ROS and Malondialdehyde, and the Activities of Antioxidant Enzymes in Aβ_(1–42)_/Cu^2+^/AA-Treated SH-SY5Y Cells

OH· could be produced by the Aβ_(1–42)_/Cu^2+^/AA system and induce oxidative damage to SH-SY5Y cells. Oxidative damage could influence the accumulation of intracellular ROS and malondialdehyde (MDA), as well as the activities of SOD and glutathione (GSH) in cells [34]. Thus, the amount of intracellular ROS and MDA, and the activities of SOD and GSH, were detected. As shown in Figure 4A,B, the level of intracellular ROS increased with the incubation of the Aβ_(1–42)_/Cu^2+^/AA mixture, while it reduced with the addition of DHPA (0.5, 4, 10 μM). To investigate whether the repression effect of DHPA on intracellular ROS production was only caused by a chemical reaction to the oxidants outside the cell, further experiments were performed. As shown in Figure 4C, DHPA solutions were added to cells after removing the Aβ_(1–42)_/Cu^2+^/AA mixture, the level of intracellular ROS was also decreased by DHPA. These results indicate that DHPA can reduce ROS production outside the cell, reduce intracellular ROS levels by its antioxidant abilities, or activate the endogenous antioxidant system in the cells. Meanwhile, the amount of MDA in SH-SY5Y was also decreased by DHPA (Figure 4D). Furthermore, the activities of SOD and GSH in SH-SY5Y cells were improved by DHPA (Figure 4E,F). These results indicated that DHPA could reduce intracellular ROS, lipid peroxidation, and increase the activities of SOD and GSH.

### 3.5. Effects of DHPA on Mitochondrial Apoptosis in Aβ_(1–42)_/Cu^2+^/AA-Induced SH-SY5Y Cells

The mitochondrion is the sensitive organelle to oxidative damage, which could induce pro-apoptotic factor expression and cell apoptosis [7]. As shown in Figure 5, with the incubation of Aβ_(1–42)_/Cu^2+^/AA, the ratio of the pro-apoptotic protein Bax to the anti-apoptotic protein Bcl-2 (Bax/Bcl-2) increased, and the mitochondrial membrane potential (MMP) significantly increased, resulting in an increase of caspase-3 (cleaved) expression. When DHPA was added to the Aβ_(1–42)_/Cu^2+^/AA-treated cells, the Bax/Bcl-2 ratio decreased, the MMP reduced, and the caspase-3 (cleaved) expression decreased. These results indicate that DHPA could inhibit Aβ_(1–42)_/Cu^2+^/AA-induced cell apoptosis via the mitochondrial apoptosis pathway.

### 3.6. Effects of DHPA on the Phosphorylation of MAPKs and AKT in Aβ_(1–42)_/Cu^2+^/AA-Induced SH-SY5Y Cells

The MAPK signaling pathway plays an important role in oxidative stress-induced neuronal cell death [35]. To investigate the effects of DHPA on the MAPK pathway, the expression of Erk and P38-related proteins were studied. As shown in Figure 6, phosphorylation of Erk1/2 and P38 was significantly enhanced by Aβ_(1–42)_/Cu^2+^/AA stimulation, while DHPA markedly decreased the expressions of P-Erk and P-P38. Moreover, the expressions of P-AKT decreased with the incubation of Aβ_(1–42)_/Cu^2+^/AA but increased significantly after DHPA treatment.

### 3.7. Effects of DHPA on the Keap1/Nrf2/HO-1 Signaling Pathway in Aβ_(1–42)_/Cu^2+^/AA-Treated SH-SY5Y Cells

The Keap1/Nrf2/HO-1 signaling pathway is considered an important pathway in the endogenous antioxidant system, and it is highly sensitive to oxidative stress [10]. To verify whether DHPA activated the Keap1/Nrf2/HO-1 signaling pathway in SH-SY5Y cells, some protein expressions were detected with the WB experiment. As shown in Figure 7, the amount of detectable Keap1 increased in Aβ_(1–42)_/Cu^2+^/AA-treated SH-SY5Y cells but decreased with the DHPA addition. The expression of Nrf2 in the cytoplasm (cyto-Nrf2) slightly increased by the Aβ_(1–42)_/Cu^2+^/AA treatment, and DHPA has a minor effect on cyto-Nrf2 expression. However, Aβ_(1–42)_/Cu^2+^/AA treatment could decrease the amount of Nrf2 in the nucleus (nucl-Nrf2), while DHPA could increase the nucl-Nrf2 amount, and result in an increase of HO-1 and NQO-1 expressions.

### 3.8. Results from Molecular Docking

Molecular docking (MD) techniques are commonly used to investigate the interactions between small molecules and the protein target [36]. To investigate the interaction between DHPA and Keap1, 1X2J (PDB ID) was selected as the crystal structure of Keap1 for the MD simulation. Compared with well-studied curcumin [21], the binding energy for the DHPA–Keap1 complex (−9.1 kcal/mol) is even lower than the one of the curcumin–Keap1 complex (−8.8 kcal/mol) (Table 1), which indicated higher binding stability of the DHPA–Keap1 complex than the curcumin–Keap1 complex [36]. At the same time, we found that the residues, namely VAL418 (bond length: 2.2 Å), VAL465 (bond length: 2.3 Å), VAL512 (bond length: 2.0 Å), ARG415 (bond length: 2.2 Å), and GLY364 (bond length: 2.4 Å), were observed to form five hydrogen bonds between DHPA and the Keap1 protein. Moreover, VAL418 (bond length: 2.2 Å), VAL465 (bond length: 2.1 Å), VAL512 (bond length: 2.5 Å), and ARG415 (bond length: 2.3 Å) were observed to form four hydrogen bonds with curcumin and the Keap1 receptor (Figure 8C). In addition, both DHPA and curcumin can form a hydrophobic Pi-Alkyl interaction between amino acid residues ALA366 and ALA556 and Keap1, but DHPA can also form a Pi–Pi stacked with Keap1 and TYR334 (Figure 8C,D), which may also be one of the reasons why the DHPA–Keap1 complex is more stable.

## 4. Discussion

According to the experimental results, we believe that DHPA has an inhibition effect on Aβ/Cu^2+^/AA-induced SH-SY5Y cell apoptosis, which is mainly caused by anti-oxidation activity (Figure 9). In the brains of AD patients, aggregated Aβ and an excess amount of Cu (~0.4 mM) were found in the senile plaque [37,38]. Moreover, research shows that toxic Aβ aggregates and oxidative stress, produced by the Aβ/Cu mixture/complex, play important roles in AD pathogenesis [33]. The Aβ–Cu^2+^ complex can promote the production of H_2_O_2_ by reacting with substances (e.g., ascorbic acid, AA, and O_2_) in cells [39,40,41]. In the cellular milieu, any unbounded Cu^2+^ will also react with H_2_O_2_ to produce hydroxyl radicals via the Haber–Weiss reaction [33].

The imbalance of redox homeostasis could increase ROS levels and result in cell lipid peroxidation and cell apoptosis. [42]. In the present study, the Aβ_(1–42)_/Cu^2+^/AA system produced hydroxyl radicals (Figure 3). Moreover, DHPA could inhibit hydroxyl radical production and protect the SH-SY5Y cells from oxidation damage induced by hydroxyl radicals (Figure 1). The amounts of intracellular ROS and MDA in SH-SY5Y cells, incubated with the Aβ_(1–42)_/Cu^2+^/AA system, were reduced by the DHPA treatment (Figure 4A,B). The activities of SOD and GSH increased with the DHPA treatment (Figure 4C,D). Moreover, the cell apoptosis of SH-SY5Y induced by the Aβ_(1–42)_/Cu^2+^/AA system was also inhibited by the DHPA treatment (Figure 2).

Apoptosis is a complex, precise, and protease-dependent molecular cascade process [43]. The Bcl-2 protein family is mainly located in the outer membrane of mitochondria, including pro-apoptotic members (Bax, etc.) and anti-apoptotic proteins (Bcl-2, etc.) [44]. Upon cell stimuli by oxidative stress, the ratio of Bax/Bcl-2 increased, which changed the MMP, activated caspase-3, and resulted in cell apoptosis [45]. Thus, the ratio of Bax/Bcl-2 and the expression level of cleaved caspase-3 play major roles in cell apoptosis [46]. Our experiments showed that Aβ_(1–42)_/Cu^2+^/AA stimulation increased the ratio of Bax/Bcl-2 proteins in SH-SY5Y cells, which reduced the MMP and increased the expression of cleaved caspase-3. However, the addition of DHPA provided significant protection against Aβ_(1–42)_/Cu^2+^/AA-induced damage by downregulating the ratio of Bax/Bcl-2 and cleaved caspase-3, alleviating MMP loss (Figure 5). These results indicate that DHPA could protect the SH-SY5Y cells from Aβ_(1–42)_/Cu^2+^/AA-induced cell apoptosis via the mitochondrial apoptotic pathway.

PI3K/Akt is an important regulatory pathway of cells; it is involved in regulating cell growth, metabolism, apoptosis, etc. [47]. Akt can be activated by phosphorylation and subsequently activate multiple downstream targets to enhance cell survival. In this study, DHPA can significantly increase the expression of the P-Akt protein (Figure 6C), suggesting that DHPA protects SH-SY5Y cells from Aβ_(1–42)_/Cu^2+^/AA-induced oxidative damage by inhibiting ROS accumulation, activating the PI3K/Akt pathway, and regulating the expressions of apoptosis-related proteins.

Mitogen-activated protein kinases (MAPKs) include JNK, P38, and ERK, among which, P38 can be stimulated by a variety of cytokines to produce corresponding effects and participate in regulating cell proliferation, differentiation, migration, invasion, and death [35]. In AD brains, phosphorylated (active) P38 (P-P38) levels are increased [48]. Moreover, many studies have shown a relationship between the activation of the P38/Erk pathway and programmed cell death [49,50]. In our study, DHPA significantly enhanced the expressions of P-P38 and P-Erk proteins (Figure 6B,D). Moreover, activated P38 induces Bax translocation and mediates the activation of caspases family proteins, which induce mitochondrial apoptosis. At the same time, PI3K/Akt and MAPKs regulate Keap1/Nrf2/HO-1 signaling pathway activation and promote nuclear translocation of Nrf2 [51].

The Keap1/Nrf2/HO-1 signaling pathway is considered the “switch” for the endogenous antioxidant system. Under normal physiological conditions, the Nrf2 signaling pathway is negatively regulated by the binding of Keap1 to the Nrf2 protein [52,53]. When the cells are stimulated with oxidative stress, Nrf2 is activated, separates from Keap1, is translocated from the cytoplasm to the nucleus, and induces the expressions of NQO1 and HO-1 [19,54]. In our study, DHPA could significantly activate Nrf2 in a dose-dependent manner and increase the expressions of NQO1 and HO-1 (Figure 7), which resulted in protective activity against oxidative stress induced by Aβ_(1–42)_/Cu^2+^/AA in SH-SY5Y cells. The molecular mechanism of the activation effect of DHPA on the Keap1/Nrf2/HO-1 signaling pathway was investigated with MD. MD could provide the binding modes and affinities between the compounds and proteins [55]. As reported in another study, curcumin could bind to the Keap1 protein to separate the binding between Keap1 and Nrf2, promote the entrance of Nrf2 to the nucleus, and activate the antioxidant system [21]. In the present study, the binding of Keap1 to curcumin and DHPA was detected. Moreover, the results show that DHPA, similar to curcumin, could bind to Keap1 via hydrogen bonds (Figure 8) (Table 1); the binding could promote the separation of Nrf2 to Keap1, which could result in the activation of the Keap1/Nrf2/HO-1 signaling pathway.

## 5. Conclusions

To summarize, DHPA can be considered an inhibitor for Aβ_(1–42)_/Cu^2+^/AA-induced oxidative damage by scavenging OH, regulating mitochondria apoptosis, and activating the Keap1/Nrf2/HO-1 signaling pathway. Our results indicate that DHPA could decrease Aβ_(1–42)_/Cu^2+^/AA-induced OH production, intracellular ROS accumulation, and MDA production. It could also repress the ratio of Bax/Bcl-2, MMP increase, activated caspase-3 expression, and the phosphorylation of Erk1/2 and P38. Furthermore, DHPA could increase the expression of P-AKT and bind with Keap1 to promote the separation of Nrf2 to Keap1 and activate the Keap1/Nrf2/HO-1 signaling pathway to increase the expressions of HO-1, NQO1, GSH, and SOD. Therefore, the results suggest that DHPA has high antioxidative abilities and might be considered a potential drug for AD.

## Figures and Tables

**Figure 1 antioxidants-11-01794-f001:**
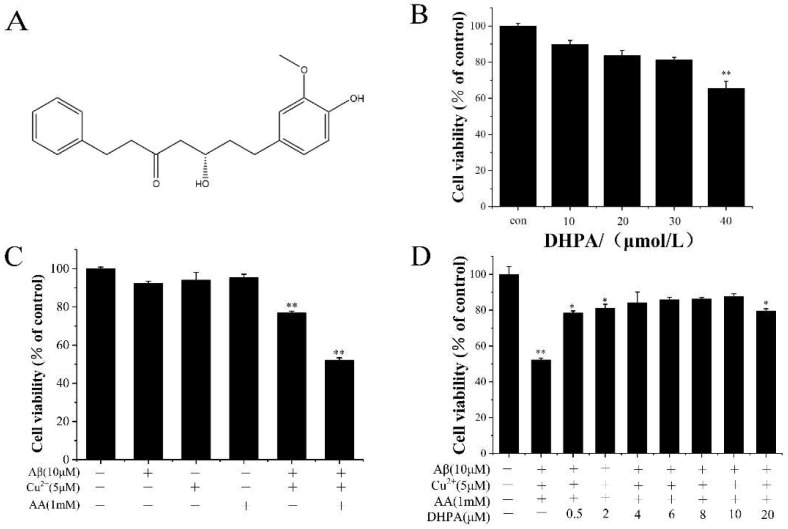
Molecular structure of DHPA (**A**); cell viability of different solution-treated SH-SY5Y cells (**B**–**D**). * *p* < 0.05 and ** *p* < 0.01 compared to the control group.

**Figure 2 antioxidants-11-01794-f002:**
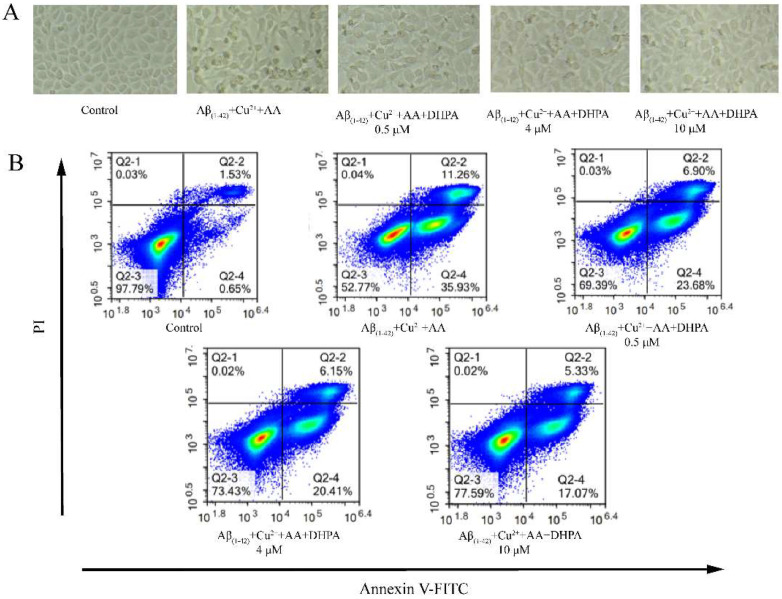
The effects of DHPA on the cell apoptosis of Aβ_(1–42)_/Cu^2+^/AA-induced SH-SY5Y cells. Cell morphology (**A**). Apoptosis rate detected with flow cytometry (**B**).

**Figure 3 antioxidants-11-01794-f003:**
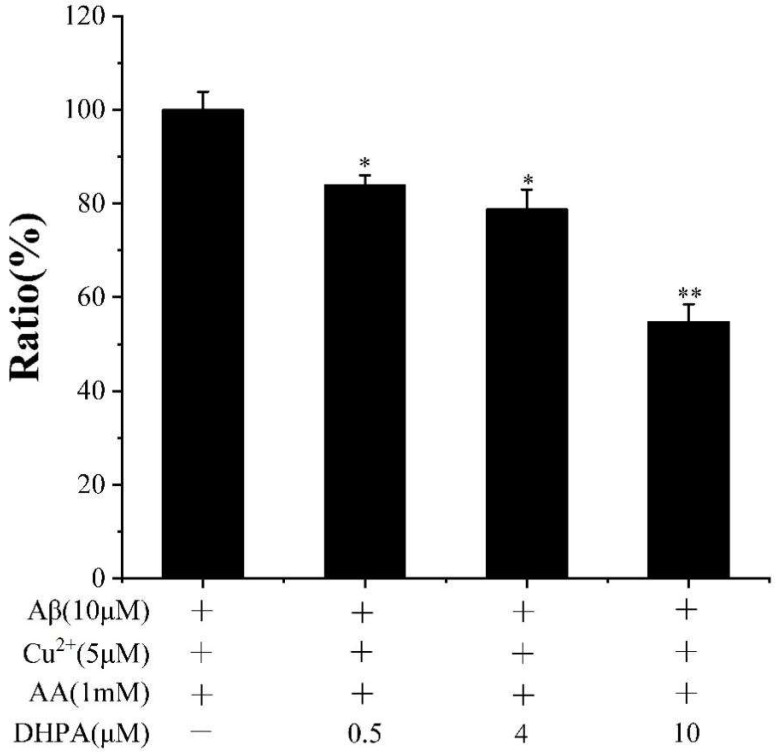
CCA fluorescence ratio of Aβ_(1–42)_/Cu^2+^/AA solutions in the absence or presence of different concentrations of DHPA. * *p* < 0.05 and ** *p* < 0.01 compared to values obtained in the Aβ_(1–42)_/Cu^2+^/AA solution.

**Figure 4 antioxidants-11-01794-f004:**
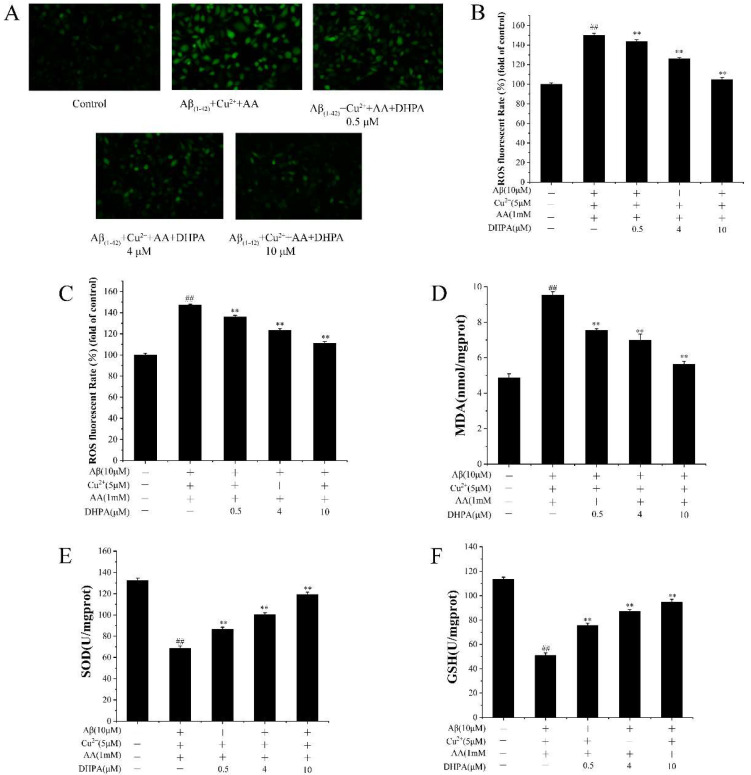
The effects of DHPA on the related antioxidant indices in SH-SY5Y cells. Intracellular ROS fluorescence images (**A**), intracellular ROS production (**B**,**C**), the content of MDA (**D**), and the activities of SOD (**E**) and GSH (**F**). For (**C**), the cells were pre-treated with Aβ/Cu^2+^/AA for 6 h, and washed afterward, with different concentrations of DHPA added. For the other experiments, Aβ/Cu^2+^/AA and DHPA were mixed for cell incubation. ## *p* < 0.01 compared to the control group, ** *p* < 0.01 compared to the Aβ/Cu^2+^/AA-treated group.

**Figure 5 antioxidants-11-01794-f005:**
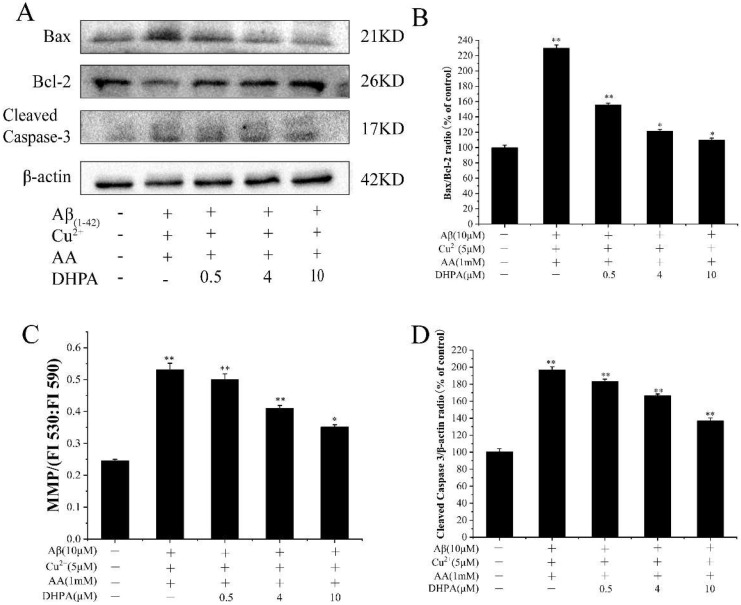
Effects of DHPA on the mitochondrial apoptosis pathway in Aβ_(1–42)_/Cu^2+^/AA-induced SH-SY5Y cells. (**A**) Representative Western blot were shown for Bax, Bcl-2, and Cleaved Caspase-3 proteins. Bax/Bcl-2 (**B**), cleaved caspase-3 (**D**), and protein expressions were detected by Western blot analysis. MMP (**C**) of SH-SY5Y cells treated with the Aβ_(1–42)_/Cu^2+^/AA mixture in the absence or presence of different concentrations of DHPA. * *p* < 0.05 and ** *p* < 0.01 compared to the control group.

**Figure 6 antioxidants-11-01794-f006:**
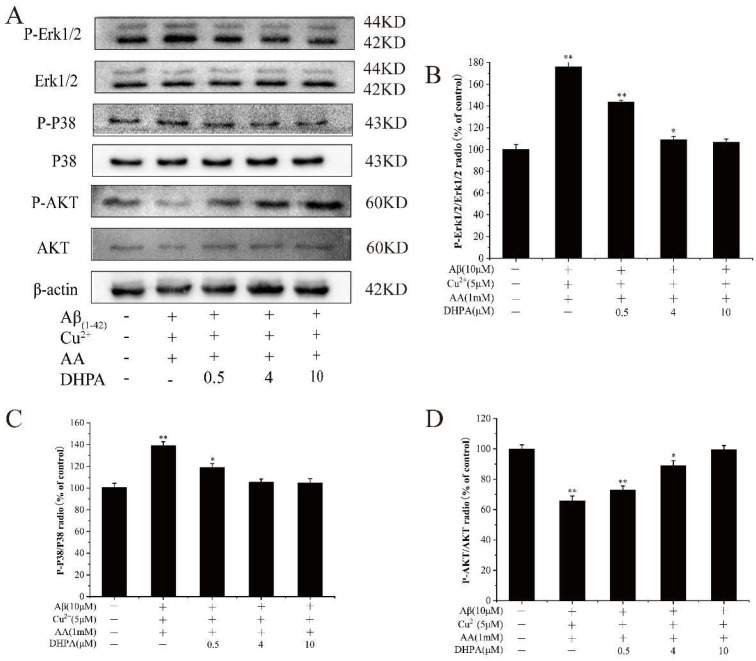
Effects of DHPA on the phosphorylation of MAPKs and AKT in Aβ_(1–42)_/Cu^2+^/AA-induced SH-SY5Y cells. (**A**) Representative Western blot were shown for P-Erk1/2, Erk1/2, P-P38, P38, P-AKT and AKT proteins. P-Erk1/2 (**B**), P-P38 (**C**), and P-AKT (**D**) protein expressions were detected by WB analysis. * *p* < 0.05 and ** *p* < 0.01 compared to the control values.

**Figure 7 antioxidants-11-01794-f007:**
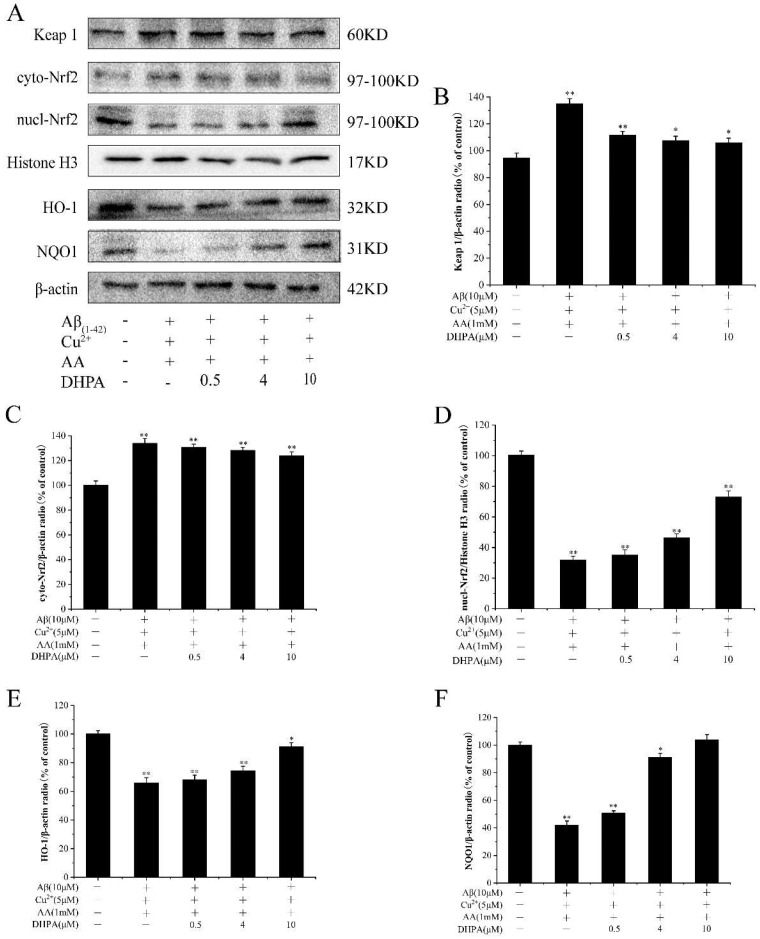
Effects of DHPA on the Keap1/Nrf2/HO-1 signaling pathway in Aβ_(1–42)_/Cu^2+^/AA-induced SH-SY5Y cells. (**A**) Representative Western blot were shown for Keap1, cyto-Nrf2, Nucl-Nrf2, Histone H3, HO-1 and NQO1 proteins. Keap1 (**B**), cyto-Nrf2 (**C**), Nucl-Nrf2 (**D**), HO-1 (**E**), and NQO1 (**F**) protein expressions were evaluated by the WB analysis. * *p* < 0.05 and ** *p* < 0.01 compared to control values.

**Figure 8 antioxidants-11-01794-f008:**
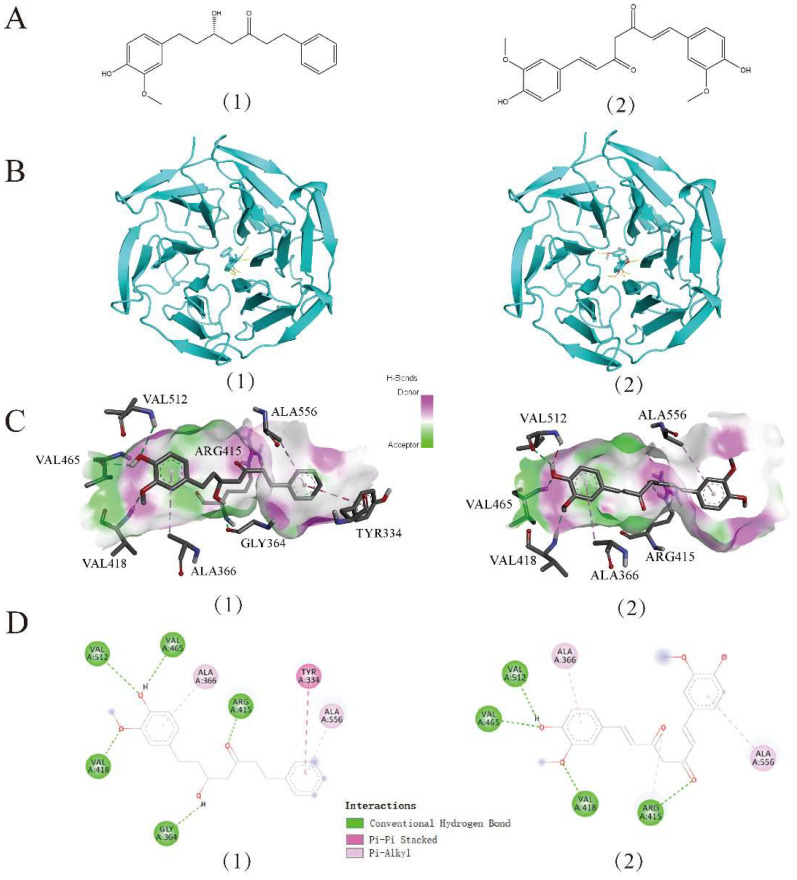
Docking between Keap1 (1X2J) and two compounds, DHPA (1) and curcumin (2). Molecular structure diagram of DHPA and curcumin (**A**). Global graph of docking results (**B**). The best binding pose, site view, and interaction graph between DHPA, curcumin, and active sites of Keap1 (**C**,**D**).

**Figure 9 antioxidants-11-01794-f009:**
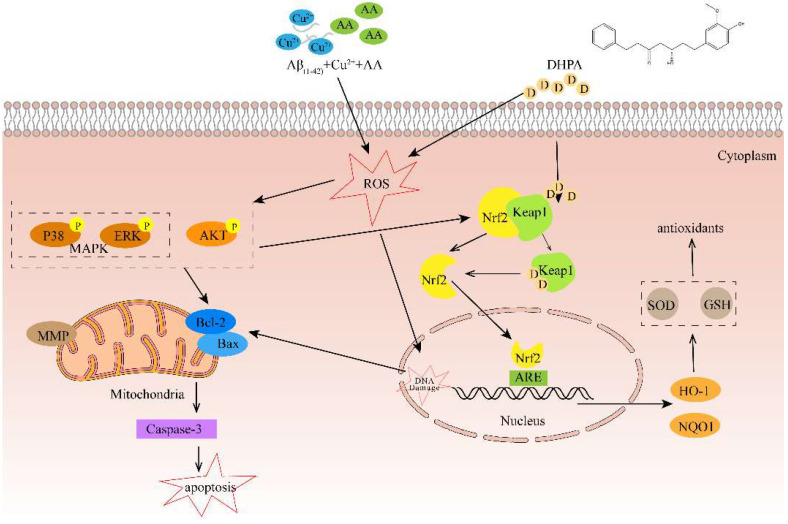
Schematic diagram of the mechanisms of DHPA protecting SH-SY5Y cells from apoptosis induced by Aβ_(1–42)_/Cu^2+^/AA.

**Table 1 antioxidants-11-01794-t001:** Docking information of the interaction of Keap1 and small molecules.

Compound	Binding Energy (kcal/mol)	Hydrophobic Interactions	Hydrogen Bonding
DHPA	−9.1	TYR334, ALA366, ALA556	VAL418, VAL465, VAL512, ARG415, GLY364
Curcumin	−8.8	ALA366, ALA556	VAL418, VAL465, VAL512, ARG415

## Data Availability

All of the data is contained within the article.

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
