# Peer review of "DHPA Protects SH-SY5Y Cells from Oxidative Stress-Induced Apoptosis via Mitochondria Apoptosis and the Keap1/Nrf2/HO-1 Signaling Pathway"

_antioxidants, 2022, doi:10.3390/antiox11091794_

Round 1

Reviewer 1 Report

This paper reports antioxidant effects of DHPA in SH-SY5Y cells treated with Aβ(1-42)/Cu2+/AA.

One important correction is necessary in the Introduction: Line 46 "Mitochondria are one of the most important target for excessed ROS" should be "Mitochondria are one of the most important sources of ROS"

English needs correcting, e.g. Line 86 "has the significant" should be "has significant"

The in vitro endpoints measured are standard markers of oxidative stress effects. There is no clear discussion of the in silico Keap1 binding data - do the authors propose that DHPA acts as an antioxidant by binding to Keap1? DHPA seems to be structurally and functionally very similar to curcumin, a well- characterised antioxidant. So nothing really new here.

Reviewer 2 Report

In this manuscript Cai and colleagues have investigated the potential protective role of DHPA against neuronal toxicity caused by ROS associated with AD, using as model the neuroblastoma cells SH-SY5Y. Although the hypothesis would present interest to the specific scientific field, there are a few conceptualization issues I am having with the presented manuscript and I believe they should be stated clearly since it could lead to the misinterpretation of the entire story. Please find below my specific comments.

Major concerns:

-it is unclear to me why the authors chose to use monomeric, rather than oligomeric or fibrillated Ab since it is well know these are the toxic forms associated with AD, as the authors themselves have represented it in figure 9

-also the authors did not consider (it was not clear to me from the main text) the fact that DHPA could chemically react to the oxidants outside the cell, and therefore diminish the effect they have from the beginning; to exclude this hypothesis additional experiments are required: first adding the Ab/Cu/AA mixture to cells, allow them to generate ROS and wash them out afterwards adding the antioxidant DHPA to properly evaluate its intracellular capacity for ROS scavenging

-the authors should detail how the experiments were performed and how the controls were designed to each figure legend (eg mention the carrier solution for the added chemicals and if the controls were performed including the carrier solutions only, otherwise the effect of the chemicals could be assigned to the carrier, such as DMSO, and the results would differ considerably)

-based on the images in figure 2A the cells presented here are overconfluent for this particular cell line, therefore the authors should explain if the same confluency was used throughout the entire manuscript or not

Minor concerns:

-I suggest the authors be moderate with using the term “mechanism” since they have only evaluated the response of several signaling pathways to chemical stress, no mechanistic details have been presented

-the methods section should be written in more detail so the presented experiments could be reproduced by the scientific community, if necessary

-a suggestion to improve the manuscript if the authors wish to: the mitochondrial membrane potential can easily be assed in parallel with cell morphology and other markers using MitoTracker -ROS (or equivalent) along with brightfield images using a fluorescent microscope, to show both ROS generation and potential changes in cell morphology on the same cell to complement the global analysis the authors have presented in the manuscript

-“in vitro” and “in silico” should be in italic throughout the text

-I recommend the use of the abbreviation of the compound in the title not the full name

-overall readability should be revised, some words are repeated extensively throughout the text

Reviewer 3 Report

Comments and suggestions

1.  Title should be revised according to your study.

2. In abstract, line 18-19, Oxidative stress in the brains of Alzheimer's disease (AD) patients is highly related with 18 the pathogenesis of AD. Please check this sentence meaning is retain. 

 3. Furthermore, DHPA could bind with Keap1 to 29 promote the separation of Nrf2 to Keap1 and activate the Keap1/Nrf2/HO-1 signaling pathway to 30 increase the expression of oxygenase-1 (HO-1), quinone oxidoreductase-1 (NQO1), glutathione 31 (GSH) and superoxide dismutase (SOD). Please check highlighted part has been written correctly.

4. Line 78-79, Alpinia officinarum Hance (Zingiberaceae Alpinia) is a traditional medicine in China, 78 and it mainly contains diarylheptanoids, flavonoids, essential oils and other bioactive 79 components [24]. Please check this name has been written correctly.

5. Materials and methods section; please check all chemicals and company name has been written correctly.

6. Figure 1 is not clear. Please provide a good figure during the revised submission.

7.  Figure 3, 4, 5 P <0.05 and * * p < 0.01 compared to values obtained in Aβ(1- 259 42)/Cu2+/AA solution. Please write the p-value consistently.

8. Figure 5. 6,7 please mention the molecular weight of protein in western blot.

9.  Figure 8. The docking image is not clear. Please provide a high-resolution image.

10. Figure 10 any copyright issue?

11. Discussion section should be updated accordingly to your results.

12. In conclusion summarize your main findings.

General comments: Lot of typos and grammatical errors throughout the manuscript that should be checked before revised submission with the help of professional expertise of this field.

Round 2

Reviewer 2 Report

The authors have addressed all my concerns, I have nothing to add. The manuscript is sutable for acceptance in the current form.

Reviewer 3 Report

The authors address my comments. So it can be accepted.